# Radiomics Driven Diffusion Weighted Imaging Sensing Strategies for Zone-Level Prostate Cancer Sensing

**DOI:** 10.3390/s20051539

**Published:** 2020-03-10

**Authors:** Chris Dulhanty, Linda Wang, Maria Cheng, Hayden Gunraj, Farzad Khalvati, Masoom A. Haider, Alexander Wong

**Affiliations:** 1Vision and Image Processing Research Group, University of Waterloo, Waterloo, ON N2L 3G1, Canada; alexander.wong@uwaterloo.ca; 2Department of Systems Design Engineering, University of Waterloo, Waterloo, ON N2L 3G1, Canada; maria.cheng@edu.uwaterloo.ca; 3Department of Mechanical and Mechatronics Engineering, University of Waterloo, Waterloo, ON N2L 3G1, Canada; hayden.gunraj@uwaterloo.ca; 4Lunenfeld-Tanenbaum Research Institute, Sinai Health System, Toronto, ON M5G 1X5, Canada; farzad.khalvati@utoronto.ca (F.K.); m.haider@utoronto.ca (M.A.H.); 5Waterloo Artificial Intelligence Institute, University of Waterloo, Waterloo, ON N2L 3G1, Canada

**Keywords:** prostate cancer sensing, zone-level sensing, radiomics, discovery radiomics

## Abstract

Prostate cancer is the most commonly diagnosed cancer in North American men; however, prognosis is relatively good given early diagnosis. This motivates the need for fast and reliable prostate cancer sensing. Diffusion weighted imaging (DWI) has gained traction in recent years as a fast non-invasive approach to cancer sensing. The most commonly used DWI sensing modality currently is apparent diffusion coefficient (ADC) imaging, with the recently introduced computed high-b value diffusion weighted imaging (CHB-DWI) showing considerable promise for cancer sensing. In this study, we investigate the efficacy of ADC and CHB-DWI sensing modalities when applied to zone-level prostate cancer sensing by introducing several radiomics driven zone-level prostate cancer sensing strategies geared around hand-engineered radiomic sequences from DWI sensing (which we term as Zone-X sensing strategies). Furthermore, we also propose Zone-DR, a discovery radiomics approach based on zone-level deep radiomic sequencer discovery that discover radiomic sequences directly for radiomics driven sensing. Experimental results using 12,466 pathology-verified zones obtained through the different DWI sensing modalities of 101 patients showed that: (i) the introduced Zone-X and Zone-DR radiomics driven sensing strategies significantly outperformed the traditional clinical heuristics driven strategy in terms of AUC, (ii) the introduced Zone-DR and Zone-SVM strategies achieved the highest sensitivity and specificity, respectively for ADC amongst the tested radiomics driven strategies, (iii) the introduced Zone-DR and Zone-LR strategies achieved the highest sensitivities for CHB-DWI amongst the tested radiomics driven strategies, and (iv) the introduced Zone-DR, Zone-LR, and Zone-SVM strategies achieved the highest specificities for CHB-DWI amongst the tested radiomics driven strategies. Furthermore, the results showed that the trade-off between sensitivity and specificity can be optimized based on the particular clinical scenario we wish to employ radiomic driven DWI prostate cancer sensing strategies for, such as clinical screening versus surgical planning. Finally, we investigate the critical regions within sensing data that led to a given radiomic sequence generated by a Zone-DR sequencer using an explainability method to get a deeper understanding on the biomarkers important for zone-level cancer sensing.

## 1. Introduction

Prostate cancer is the most commonly diagnosed type of cancer in North American men, excluding non-melanoma skin cancer, and is one of the leading causes of cancerous death in North American men [1,2], accounting for an estimated 164,690 new cases and 29,430 deaths in the United States in 2018. However, prognosis is relatively good given sufficiently early detection, motivating the need for fast and reliable cancer screening methods.

Diffusion weighted imaging (DWI) is a magnetic resonance imaging (MRI) technique that is gaining traction as a noninvasive method for prostate cancer sensing. Several DWI sensing modalities have been proposed to better differentiate between healthy and cancerous tissue. Currently, apparent diffusion coefficient (ADC) imaging are the most commonly used DWI sensing modality for cancer sensing. Computed high-b value diffusion weighted imaging (CHB-DWI) is a different DWI sensing modality that has shown improved distinction between healthy and cancerous tissue [3,4], but this method has not been widely adopted.

The current clinical practice is a clinical heuristics driven strategy, where a heuristics based threshold derived from the observations made in past clinical studies is leveraged to detect areas that likely contain cancer [5]. Although this approach leverages important clinical findings, it lacks the ability to characterize more complex spatial traits such as textural or morphological traits that can differentiate between healthy and cancerous prostate tissue, and may be limited in its ability to sense prostate cancer.

Radiomics driven cancer sensing methods have been shown to be a promising prognostic tool [6,7,8], but rely on predefined and hand-engineered quantitative radiomic features. Recently, *discovery radiomics* [9] was introduced to uncover abstract radiomic features, directly from the wealth of sensing data, that capture highly unique tumour traits and characteristics beyond what can be captured using existing radiomics driven sensing approaches.

Previous studies have focused on slice-level prostate cancer sensing. However, it can potentially be very beneficial to grade tissue on a smaller scale at a zone level as tumors in different zones can have different characteristics. Furthermore, zone-level cancer sensing can also help isolate the precise location of cancer for more targeted biopsy and treatment.

In this study, we investigate comprehensively the efficacy of two DWI sensing modalities, ADC and CHB-DWI, for radiomics driven zone-level prostate cancer sensing. In order to do this investigation, we introduced several radiomics driven strategies (which we will term as Zone-X sensing strategies) for zone-level DWI prostate cancer sensing geared around hand-engineered radiomic sequences that have not been previously introduced in past literature for the purpose of zone-based prostate cancer sensing, which is an important contribution of this study. Furthermore, for another important contribution of this study, we also introduce Zone-DR, a discovery radiomics driven strategy based on zone-level deep radiomic sequencer discovery. Demonstrating the efficacy of such radiomics driven strategies that leverage DWI sensing modalities can hopefully lead to their widespread adoption for improved zone-based prostate cancer sensing, as using a non-invasive sensing method can reduce the number of negative surgical biopsies and improve the detection of tumors.

## 2. Diffusion Weighted Imaging for Cancer Sensing

In this study, two DWI sensing modalities are leveraged: i) apparent diffusion coefficient (ADC) imaging, and ii) computed high-b diffusion-weighted imaging (CHB-DWI). The details of these DWI sensing modalities are specified below, and illustrative examples are displayed in Figure 1.

### 2.1. Apparent Diffusion Coefficient Imaging

DWI measures the sensitivity of tissue to the Brownian motion of water molecules by applying pairs of opposing gradient pulses on either side of refocusing pulses in spin-echo sequences [10]. The duration and amplitude of these gradient pulses are represented by a *b*-value, and the diffusion-weighted signal *S* decays exponentially with respect to:(1)$$S=S_{0}e^{-bD}$$

In (Equation 1), $S_{0}$ is the signal intensity with no diffusion-weighting (b=0) and *D* is the ADC. DWI is typically performed for different *b*-values, allowing ADC images to be computed by fitting the signal curve with parameters $S_{0}$ and *D* via least-squares or maximum likelihood methods [11]. Cancerous tissue presents with a lower ADC relative to surrounding healthy tissue, allowing it to be identified in estimated ADC images [10].

### 2.2. Computed High b-Value Diffusion Weighted Imaging

It has been shown that DWI using high *b*-values (i.e., *b*-values greater than 1000 s/mm${}^{2}$) allows for improved distinction between healthy and cancerous tissue due to the higher signal intensities exhibited by cancerous tissue [3,4]. However, due to hardware limitations, high b-value DWI is difficult to achieve at a high enough signal-to-noise ratio to provide quality acquisitions for diagnostic purposes. To attain a higher signal-to-noise ratio, computed high *b*-value DWI was introduced, where a set of low *b*-value acquisitions are used to estimate higher *b*-value acquisitions [12].

For a diffusion weighted signal $S_{i}$ at a *b*-value of $b_{i}$, the general equation is:(2)$$S_{i}=S_{\alpha }e^{-(b_{i}-b_{\alpha })D}$$
where $S_{\alpha }$ is the reference signal at a *b*-value of $b_{\alpha }$ and *D* is the ADC. An estimate $\hat{D}$ of the ADC may be formulated as a Bayesian estimation problem, where *S* is a set of DWI measurements, *D* is the ADC, P(S|D) is the probability of *S* given *D*, and $S_{i}$ is a single DWI measurement corresponding to *b*-value $b_{i}$ [3]:(3)$$\begin{array}{cc} \hat{D} & =\underset{D}{\arg\max}\;P\left(S|D\right) \end{array}$$(4)$$\begin{array}{cc} \; & =\prod_{i}\;P\left(S_{i}|D\right) \end{array}$$

Once the ADC has been estimated, the estimate $\hat{D}$ can be used to compute signals ${\hat{S}}_{i}$ at any value of $b_{i}$ via (Equation 2):(5)$${\hat{S}}_{i}=S_{\alpha }e^{-\left(b_{i}-b_{\alpha }\right)\hat{D}}$$

## 3. Radiomics Driven Cancer Sensing

Existing radiomics driven cancer sensing methods for prostate cancer typically rely on quantitative, hand-engineered radiomic sequences derived from mono- or multi-parametric MRI. Notably, existing feature-based methods typically define features and perform grading on a per-pixel or per-region-of-interest basis.

A comprehensive review of existing radiomics driven methods for prostate cancer sensing was authored by Lemaître et al. [13]. The examined methods used hand-engineered radiomic feature sets derived from some combination of T2-weighted (T2w) imaging, dynamic contrast enhanced (DCE) imaging, diffusion-weighted imaging (DWI), and magnetic resonance spectroscopic imaging (MRSI). Additionally, a number of methods make use of apparent diffusion coefficient (ADC) imaging. For radiomic-driven cancer sensing and detection, the examined methods utilize the following classifiers, or some combination thereof: linear/quadratic discriminant analysis, logistic regression, naïve Bayes, AdaBoost, random forest, support vector machines (SVMs), relevance vector machines (RVMs), neural networks, Markov random fields, and conditional random fields.

Duda et al. [14] proposed a semi-automatic texture analysis method that combined features from various modalities, including DWI. For each modality, features were computed based on first order statistics, autocorrelation, gradients, fractals, co-occurrence matrices, and run length matrices. Litjens et al. [15] extracted various features, such as second-order statistical and Gabor features from T2w images, multi-scale blobness from ADC maps, and curve fitting and pharmacokinetic features from DCE images to perform prostate gland segmentation, cancer likelihood mapping, and cancerous region classification.

Ozer et al. [16] combined parametric images derived from DCE imaging, T2w imaging, and ADC imaging. To classify the sensing data, the use of relevance vector machines (RVMs) with a Bayesian framework was proposed, and was subsequently evaluated against the performance of SVMs with the same framework. Ozer et al. [17] later extended their work to select a threshold value for increased segmentation performance, and further compared with a representative Markov random field approach.

Artan et al. [18] engineered feature vectors from median-filtered multispectral MRI sensing data consisting of axial-oblique fast spin-echo (FSE) T2w images, echo-planar DWI, multi-echo FSE images, and DCE images. These features were used to develop a cost-sensitive SVM for automated prostate cancer grading, which showed improved accuracy over conventional SVMs. A conditional random field was then added to the cost-sensitive SVM framework, further improving localization accuracy.

Vos et al. [19] developed a fully-automated prostate cancer sensing method using a supervised two-stage classification approach. Lesion candidates were analysed via a combination of histogram analyses of axial T2w images, pharamcokinetic maps, contrast-enhanced T1w images, and ADC maps with texture-based features. Vos et al. then discriminated prostate cancers from benign abnormalities by their heterogeneity. Khalvati et al. [20] proposed a multi-parametric MRI texture feature model for radiomics driven prostate cancer sensing. The texture feature model comprises 19 low-level texture features extracted from each MRI modality, and is based on the model proposed by Peng et al. [21]. More recently, Khalvati et al. [8] extended this texture feature model to include additional MRI modalities and low-level features, as well as feature selection. Chaddad et al. [22] proposed the extraction of features from the joint intensity matrices (JIMs) of T2w imaging data and DWI data. Using Haralick texture features extracted from both the JIMs and co-occurrence matrices, random forest classifiers were used compare the utility of classification using features derived from JIMs to that of features derived from co-occurrence matrices. It was also shown that combining features from JIMs and co-occurrence matrices further improved cancer grading performance.

Apart from hand-engineered radiomic sequences for prostate cancer, these radiomic sequences can be drawn from other types of cancer sensing, such as lung cancer. Narayanan et al. [23] explored the performance of SVM on a large set of 503 features and shortlisted these features to 300 based on a feature ranking algorithm. Recently, Narayanan et al. [24] introduced a novel optimization method for selecting features from computed tomography (CT) and chest radiographs (CRs) for clustering and classification of lung cancer. The proposed method adapts the feature selection process based on the task in hand.

In addition to hand-engineered radiomic sequences, the notion of discovery radiomics has previously been proposed to prostate cancer sensing as well as other problems in radiomics driven sensing [9,25,26,27,28]. Chung et al. [25] proposed a fully-automated discovery radiomics system for sensing prostate lesion candidates using multi-parametric MRI (MP-MRI). Radiomics features were extracted using a discovered radiomics sequencer consisting of 17 convolutional sequencing layers and 2 fully-connected sequencing layers, and the discovered sequences were evaluated against the hand-engineered radiomic sequences described in [8,21]. More recently, Chung et al. [26] introduced a Layered Random Projection (LaRP) sequencer, which was evaluated using the same framework as [25]. Karimi et al. [27] extended the methods of [25,26] to use a mixture of convolutional sequencers in order to mitigate the effects of class imbalance. This approach was shown to improve specificity and accuracy when compared to [25].

## 4. Materials and Methods

### 4.1. Image Data and Pre-Processing

DWI sensing data (including ADC imaging and CHB-DWI imaging modalities) of 101 patients was acquired using a Philips Achieva 3.0 T MRI machine at Sunnybrook Health Sciences Centre in Toronto, Ontario. Institutional research ethics board approval and written informed consent was waived by the Research Ethics Board of Sunnybrook Health Sciences Centre.

The 3D MRI volume of each patient was divided into 2D slices along the coronal plane, resulting in 18 to 34 slices per patient. Each slice was manually annotated by an expert radiologist, producing a segmentation map of the prostate and its 10 anatomical zones [29]. An example of how the prostate is split into zones is shown in Figure 2. Each zone was labeled with a PI-RADS score between 1 and 5, representing a *very low*, *low*, *intermediate*, *high* or *very high* chance that clinically significant cancer was highly likely to be present [30]. 72 patients had one or more zone with a PI-RADS score of 3 or above, and these individuals were referred to obtain biopsies of the potentially cancerous regions. Samples were collected and a histopathological assessment was performed by a trained pathologist using the Gleason grading system [31] to assess the prognosis of patients. In total, 42 zones received a Gleason score of six, 78 received a score of seven, 12 received a score of eight and three received a score of nine, indicating the presence of a cancerous tumor (positive-grade) in 41 patients.

Prostate zones were extracted from the image slices based on a zone-level map, and cropping the area corresponding to each zone. For ADC, pixels not contained in a zone were masked with the value of 3,949, the maximum ADC value, as lower ADC values indicate a higher likelihood of cancer. For CHB-DWI, pixels not contained in a zone were masked with zero values. This process resulted in extracted zones of varying dimensions, so all examples were resized to 32 by 32 pixels using bilinear interpolation.

In total, 12,466 prostate zones were extracted and split into a dataset of 12,361 negative-grade and 135 positive-grade, pathology-verified zones. Five stratified splits of the dataset were calculated at the patient-level such that the percentage of positive-grade zones for each split was preserved. To obtain these splits, five folds were randomly selected at the patient-level for patients with no presence of a cancerous tumor and patients with presence of a cancerous tumor verified by biopsies. This resulted in four groups of 20 patients and one group of 21 patients. Each group has at least 8 patients with presence of a cancerous tumor. These five splits were used to perform 5-fold cross validation.

### 4.2. Clinical Heuristics Driven Strategy

To act as a baseline reference for comparison, the widely-used clinical heuristics driven strategy was evaluated in this study for the purpose of zone-level prostate cancer sensing. More specifically [32], for the heuristics driven strategy evaluated in this study, the following clinical heuristics presented in past studies [3,4] were used:When leveraging ADC sensing, any zone with ADC value less than 1000 s/mm${}^{2}$ are considered cancerous.When leveraging CHB-DWI sensing, any zone with CHB-DWI values greater than 1000 s/mm${}^{2}$ are considered cancerous.

The estimated probability of cancer in each ADC zone was computed as,
(6)$$1-\frac{\min\;\mathrm{ADC}\;\mathrm{value}\;\mathrm{of}\;\mathrm{zone}}{M_{ADC}}$$
where $M_{ADC}$ is the maximum ADC value. The estimated probability of cancer in each CHB-DWI zone was computed as,
(7)$$\frac{\max\;\mathrm{CHB}-\mathrm{DWI}\;\mathrm{value}\;\mathrm{of}\;\mathrm{zone}}{M_{CHB}}$$
where *M* is the maximum CHB-DWI value.

With these probabilities, a receiver operating characteristic curve is constructed to determine the area under the curve and the optimal threshold value for each sensing modality.

### 4.3. Zone-X: Radiomics Driven Sensing Strategies

In this study, we introduce several radiomics driven DWI sensing strategies (which we refer to as Zone-X sensing strategies) in the form of support vector machines (SVM), logistic regression, and random forest techniques using hand-engineered radiomic sequences for the task of zone-level cancer sensing. It is important to note that the Zone-X sensing strategies explored in this study have not been previously introduced in past literature for the purpose of zone-based DWI prostate cancer sensing, and is thus an important contribution of this study. The Zone-X sensing strategies investigated in this study are briefly discussed below.

#### 4.3.1. Support Vector Machines (SVM)

Initially proposed and optimized by Vapnik et al. [33], SVM is a supervised learning classification algorithm that performs binary classification by determining the optimal hyperplane between two classes via the maximal margin of separation. The extended use of the kernel trick for computation and soft margins allow for creation of nonlinear classifiers that can be applied in real-world applications.

#### 4.3.2. Random Forest

Proposed by Ho in 1995, random forest is an implementation of decision tree classifiers that prevents overfitting [34]. Ho suggested that subsets of the feature space could be used to generate decision trees and form an ensemble, to obtain a more accurate prediction. Random forest advantages include minimal tuning of hyperparameters and the ability to view the relative importance of input features.

#### 4.3.3. Logistic Regression

A standard classification technique, logistic regression is the estimation of the parameters that will fit the logistic model to the data. Algorithms to do so include gradient descent or maximum-likelihood estimation. However, the logistic regression technique is limited to performing linear classification, and are prone to overfitting.

In this study, a zone-level hand-engineered radiomic sequence based on Khalvati et al. [8] is modified for the purpose of zone-level prostate cancer sensing and leveraged in the proposed Zone-X sensing strategies. This zone-level radiomic sequence consists of four first-order statistical features (mean, standard deviation, skewness, kurtosis), 18 Haralick features in four directions, Kirsch features in eight directions, and Gabor features in four directions and three scales within a zone for a 96-dimensional zone-level radiomic sequence.

### 4.4. Zone-DR: Discovery Radiomics Driven Sensing

In this study, a two-part discovery radiomics strategy (which we will term as Zone-DR) for discovering zone-level radiomic sequencer discovery for the purpose of zone-level prostate cancer sensing is also introduced. An overview of the proposed Zone-DR approach is shown in Figure 3. The two-part discovery radiomics strategy comprises of: (i) machine-driven sequencer design discovery to discover the radiomic sequencer design, and (ii) data-driven sequencer parameter discovery to discover the parameters of the radiomic sequencer. This discovery radiomics strategy was leveraged to discover zone-level radiomic sequencers for both ADC and CHB-DWI sensing modalities using a wealth of zone-level sensing data of the prostate gland across an archive of patient cases. Using the final discovered radiomic sequencers, zone-level sensing acquisitions for current patient is passed through the discovered radiomic sequencers to produce radiomic sequences that characterize the tissue phenotype within the input zone based on information captured in the respective sensing modality. The generated radiomic sequence can then be used to grade the input prostate zone. The details of the different steps of the proposed Zone-DR driven sensing strategy is described in detail below.

#### 4.4.1. Machine-Driven Radiomic Sequencer Design Discovery

The radiomic sequencers leveraged in this study for the proposed Zone-DR are highly compact deep convolutional radiomic sequencers, each designed and discovered via discovery radiomics specifically to generate radiomic sequences given zone-level sensing data captured from a specific sensing modality (ADC or CHB-DWI) that quantitatively characterize tissue phenotype associated with prostate cancer for zone-level cancer grading. Since this is the first study on zone-based prostate cancer sensing via discovery radiomics, it is important to investigate and identify the appropriate zone-level radiomic sequencer design for the sensing modalities being explored.

To achieve the goal of identifying the appropriate zone-level radiomic sequencer design, we employ a machine-driven radiomic sequencer design discovery strategy, where we leverage generative synthesis [35] based on a deep convolutional-based radiomic sequencer design prototype to discover the best deep radiomic sequencer design for zone-based grading based on the area under the curve (AUC) metric as the universal performance function for each modality. More specifically, the deep convolutional-based radiomic sequencer design prototype leveraged in this study for machine-driven radiomic sequencer design discovery strategy was designed with sequencer efficiency and generalization capabilities in mind and inspired by [36], where a kernel size of 3 was leveraged in the sequencer design for computational efficiency while achieving strong grading performance, the flexibility for maxpool operations to be added at any depth of the sequencer design, and finally the incorporation of a sequence operation at the end of the sequencer design to generating the output radiomic sequences given the input zone-level sensing data.

The final discovered zone-level deep radiomic sequencer designs that achieved the best AUC performance are shown in Figure 4. A number of observations can be made about the discovered radiomic sequencer designs. First, it can be observed that the deep radiomic sequencer designs discovered via the machine-driven radiomic sequencer design strategy exhibit noticeable architectural heterogeneity as they are tailored specifically for zone-level prostate cancer sensing. Second, it can also be observed that the discovered ADC sequencer design is noticeably less complex than the CHB-DWI sequencer design discovered via the machine-driven sequencer design discovery process. This suggests that a less complex radiomic sequencer design is sufficient to characterize the tissue phenotype captured via ADC sensing modality compared to that needed for CHB-DWI sensing modality.

#### 4.4.2. Data-Driven Radiomic Sequencer Parameter Discovery

The zone-level deep radiomic sequencers designed via the aforementioned machine-driven design discovery process then undergo a radiomic sequencer discovery process to discover all the parameters of the sequencer given sensing data to capture the cancer phenotype of cancerous tissue. More specifically, an iterative adaptive gradient descent optimization method (in this study, the Adam optimizer) is leveraged with a categorical cross-entropy loss function, shown in Equation (Equation 8), to discover the parameters of the radiomic sequencers. To account for grade imbalance, grade weights of 1 and 150 are used for learning characteristics about the negative-grade and positive-grade zones, respectively.
(8)$$H_{p}\left(q\right)=-\frac{1}{N}\sum_{i=1}^{N}y_{i}\cdot log\left(p\left(y_{i}\right)\right)+\left(1-y_{i}\right)\cdot log\left(1-p\left(y_{i}\right)\right)$$

The hyperparameters that discovered the best parameters of the radiomic sequencers are shown in Table 1.

## 5. Results

To evaluate the efficacy of the introduced Zone-X and Zone-DR strategies for zone-level DWI prostate cancer sensing, an empirical five-fold performance analysis using the following performance metrics was conducted for the proposed Zone-X and Zone-DR sensing strategies, along with the baseline clinical heuristics driven sensing strategy: (i) area under the curve (AUC), (ii) sensitivity, and (iii) specificity.

To determine the optimal zone-level grading threshold for each zone-level prostate sensing strategy, denoted here as $\hat{\theta }$, we solve the following optimization problem:(9)$$\hat{\theta }=\underset{\theta }{\mathrm{argmax}}\left(\left(1-fpr\left(\theta \right)\right)+tpr\left(\theta \right)\right)$$
where fpr(θ) and tpr(θ) denotes the false positive rate and true positive rate, respectively. Note that while for this study the weights on 1−fpr(θ) and tpr(θ) are equal in the above optimization formulation, one can change the weight to determine a zone-level grading threshold that favor the zone-level sensing strategies towards either higher sensitivity or higher specificity.

Figure 5 and Figure 6 present a quantitative comparison of the AUC, sensitivity, and specificity achieved using the introduced Zone-X and Zone-DR Zone-DR sensing techniques, as well as the baseline clinical heuristics driven sensing strategy, for ADC and CHB-DWI sensing modalities, respectively. Here, we will refer to the proposed Zone-X radiomics driven zone-level prostate cancer sensing strategies leveraging logistic regression, random forest, and support vector machines as Zone-LR, Zone-RF, and Zone-SVM, respectively.

One particular advantage of the Zone-DR sensing strategy is the fact that it leverages discovered radiomic sequencers where the critical regions that led to a given radiomic sequence can be visualized using explainability methods such as [37]. Therefore, in addition to the empirical results illustrating the efficacy of the introduced radiomics driven zone-level prostate cancer sensing strategies, we further visualize the critical regions leveraged by the proposed Zone-DR sensing strategy on several example CHB-DWI positive and negative samples using GSInquire, a state-of-the-art explainability method that have been demonstrated to better reflect decision-making processes when compared to other popular explainability methods [37]. The visualizations are shown in Figure 7.

## 6. Discussion

A number of observations can be made from Figure 5 and Figure 6. First, for both ADC and CHB-DWI sensing modality, the introduced Zone-X and Zone-DR prostate cancer sensing strategies outperform the clinical heuristics driven sensing strategy in terms of AUC performance metrics (e.g., clinical heuristics driven strategy achieved on average ∼0.79 while Zone-DR achieved on average ∼0.86 for ADC sensing modality). This suggests that zone-level radiomic sequences that provides better characterization of tissue traits is important for distinguishing between positive and negative zones. Second, it can be observed that the Zone-DR sensing strategy achieved the highest sensitivity compared to the other radiomics driven sensing strategies when using ADC sensing modality, while Zone-SVM strategy achieved the highest specificity compared to the other radiomics driven sensing strategies when using ADC. Third, it can be observed that the Zone-LR and Zone-DR sensing strategies achieved the highest sensitivities compared to the other radiomics driven sensing strategies when using CHB-DWI sensing modality. Fourth, it can be observed that the Zone-LR, Zone-SVM, and Zone-DR sensing strategies achieved the highest specificities compared to the other radiomics driven sensing strategies when using CHB-DWI sensing modality. Fifth and finally, it was observed the use of CHB-DWI led to higher specificity while the use of ADC led to highest sensitivity, making the choice of sensing modality useful for different clinical scenarios. For example, maximizing specificity is important for surgery for removal of prostate where you want to minimize false positive rates to avoid unnecessary surgeries. On the other hand, for cancer screening, maximizing sensitivity may be useful to avoid missing cancerous patients.

By visual inspection in Figure 1, the cancerous tissue is more apparent for the CHB-DWI prostate slice than the ADC prostate slice. In addition, Table 1 shows that CHB-DWI sensing modality is easier to leverage and the results in Figure 5 and Figure 6 indicate that CHB-DWI achieves higher accuracy, as well as a better balance between specificity and sensitivity, when compared to the design for ADC modality. These results show that the CHB-DWI sensing modality provides superior cancer tissue characterization when used within a radiomics driven sensing strategy. Based on the experimental results, it can be observed that introduced Zone-X and Zone-DR radiomics driven DWI prostate cancer sensing strategies can provide significantly improved cancer sensing performance.

As mentioned earlier, an interesting benefit of the Zone-DR sensing strategy is that the critical regions that led to a given radiomic sequence generated by the discovered radiomic sequencer can be visualized using explainability methods. Based on Figure 7, which provide visualizations of Zone-DR critical regions on both positive zones and negative zones, it can be observed that the critical regions being leveraged by Zone-DR is consistent with the markedly signal hyperintensity characteristics in CHB-DWI used by radiologists when conducting PI-RADS assessments. More specifically, for the positive examples (Figure 7a), Zone-DR focuses on regions exhibiting signal hyperintensity in CHB-DWI sensing data. When cancer is not present in a zone (Figure 7b), Zone-DR focuses on a larger region, as shown by the critical region identified by GSInquire as the driving factor for Zone-DR for the negative example.

## 7. Conclusions

In this study, we introduced and demonstrated the efficacy of several introduced radiomics driven sensing strategies (Zone-X sensing strategies) using ADC and CHB-DWI modalities for zone-level prostate cancer sensing. Furthermore, we additionally introduced Zone-DR, a discovery radiomics sensing strategy based on zone-level deep radiomic sequencer discovery that discover radiomic sequences directly for radiomics driven sensing. The introduced Zone-X and Zone-DR sensing strategies were able to achieve noticeably higher performance when compared to the clinical heuristics driven strategy with respect to AUC performance metrics. Furthermore, the experimental results showed that the trade-off between sensitivity and specificity can be optimized based on the particular clinical scenario we wish to employ radiomics driven DWI prostate cancer sensing strategies for, such as clinical screening versus surgical planning. These promising results suggest that radiomics driven DWI sensing strategies such as the proposed Zone-X sensing strategies and discovery radiomics driven DWI sensing strategies such as the proposed Zone-DR sensing strategy can potentially be a very powerful tool for aiding radiologists in zone-level prostate cancer screening.

## Figures and Tables

**Figure 1 sensors-20-01539-f001:**
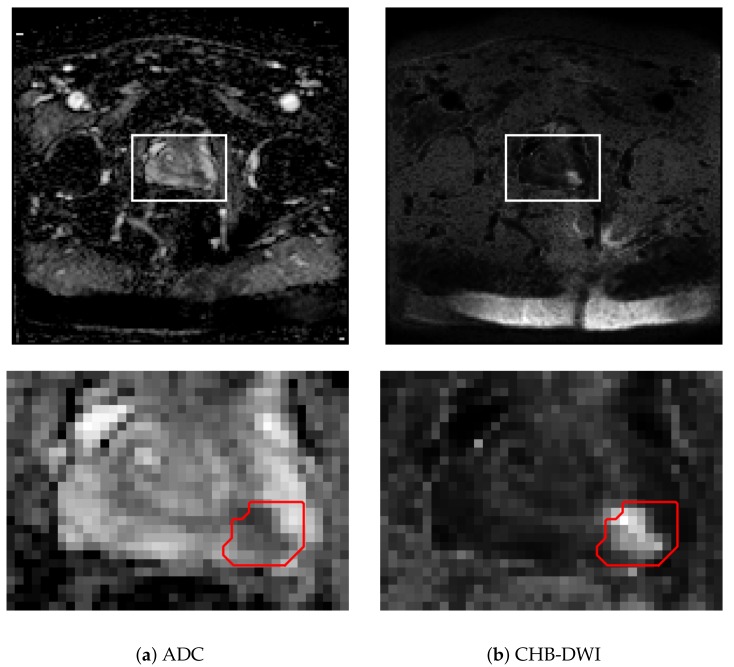
ADC (**a**) and CHB-DWI (**b**) sensing modalities for an example acquisition. Slice (top). Cropped view of the prostate (bottom). This patient has pathologically-verified prostate in zone 10 (Gleason score of 8), contoured in red.

**Figure 2 sensors-20-01539-f002:**
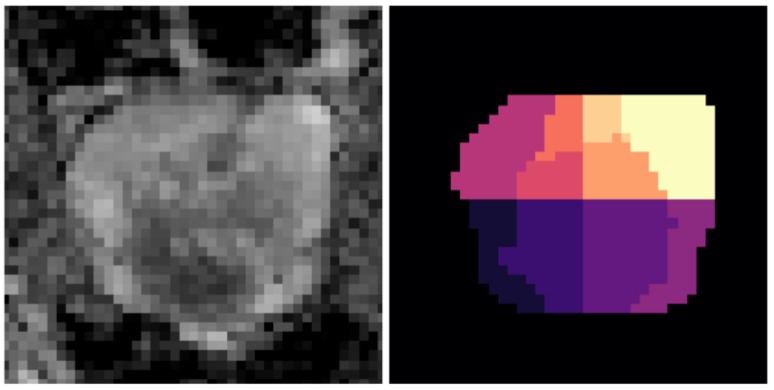
Visualization of the ten anatomical prostate zones used in the study. (left) A prostate slice, (right) the corresponding zone-level map.

**Figure 3 sensors-20-01539-f003:**
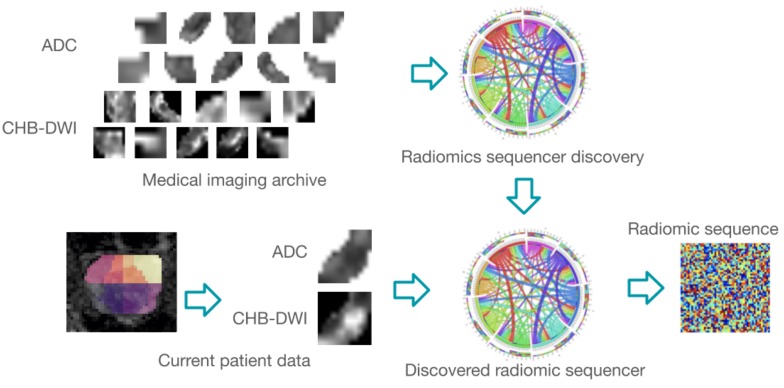
Overview of Zone-DR. For this study, radiomic sequencers were discovered via a two-part discovery radiomics strategy (machine-driven sequencer design discovery to discover the radiomic sequencer design, followed by data-driven sequencer parameter discovery to discover the parameters of the radiomic sequencer) for both ADC and CHB-DWI sensing modalities using a wealth of zone-level sensing data of the prostate gland across an archive of patient cases. Using the final discovered radiomic sequencers, zone-level sensing data of a current patient is passed through the discovered radiomic sequencers to produce radiomic sequences that characterize the tissue phenotype within the input zone based on information captured in the respective modality. The generated radiomic sequence can then be used to grade the input prostate zone.

**Figure 4 sensors-20-01539-f004:**
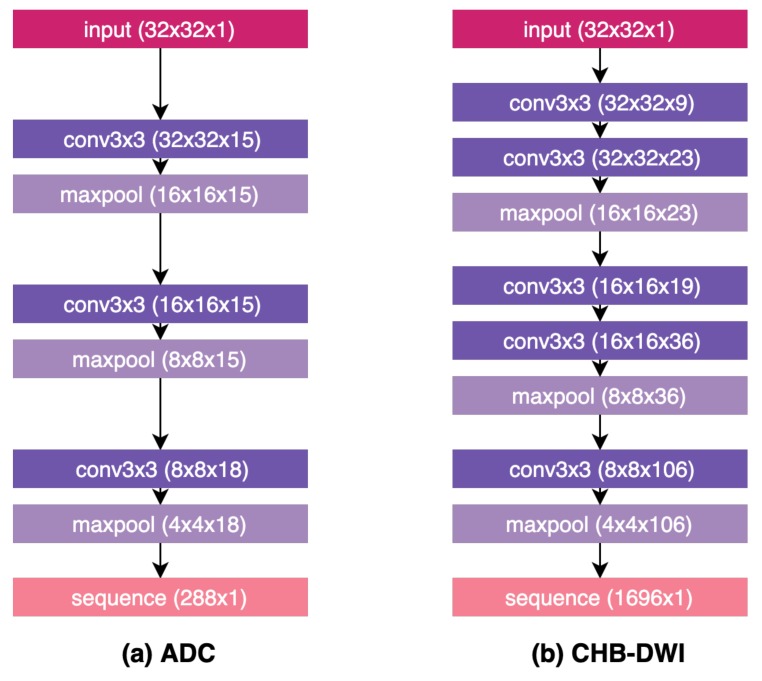
Radiomic sequencer designs discovered via a machine-driven radiomic sequencer design strategy for (**a**) ADC sensing modality and (**b**) CHB-DWI sensing modality.

**Figure 5 sensors-20-01539-f005:**
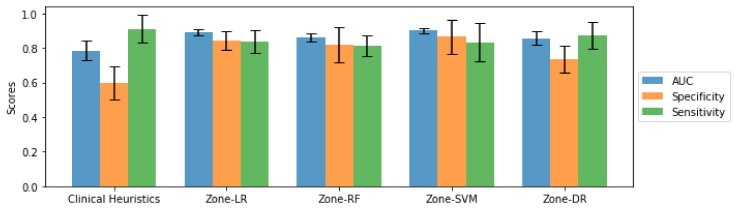
Performance bar plot (with error bars) of zone-level, pathologically-verified prostate cancer grading results for the tested zone-level prostate cancer sensing strategies using ADC, averaged over 5 folds, along with their corresponding standard deviation.

**Figure 6 sensors-20-01539-f006:**
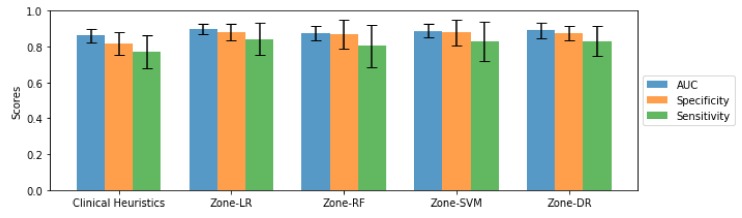
Performance bar plot (with error bars) zone-level, pathologically-verified prostate cancer grading results for the tested zone-level prostate cancer sensing strategies using CHB-DWI, averaged over 5 folds, along with their corresponding standard deviation.

**Figure 7 sensors-20-01539-f007:**
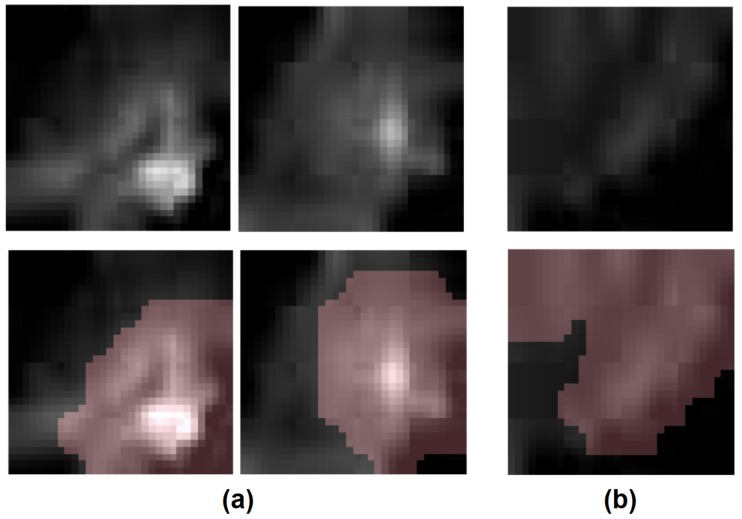
Visualization of Zone-DR critical regions on CHB-DWI sensing data using GSInquire [37]. (**a**): positive zone examples, (**b**): negative zone examples. Top: CHB-DWI sensing data, Bottom: corresponding critical region visualization with identified critical regions overlaid on their respective sensing data samples.

**Table 1 sensors-20-01539-t001:** Hyperparameters used for data-driven radiomic sequencer parameter discovery.

	ADC	CHB-DWI
batch size	219	163
learning rate	0.0002	0.002
patience	50	16

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
