# Peer review of "Radiomics Driven Diffusion Weighted Imaging Sensing Strategies for Zone-Level Prostate Cancer Sensing"

_sensors, 2020, doi:10.3390/s20051539_

Round 1
Reviewer 1 Report
Diffusion weighted imaging (DWI) is a non-invasive magnetic resonance imaging (MRI) technique for cancer sensing. Radiomics is a superior method of mining big data in medical imaging. This work investigated the efficacy of radiomics-driven DWI sensing strategies for zone-level prostate cancer sensing. Two DWI sensing modalities (ADC and CHB-DWI) with several different radiomics approaches (Zone-X and Zone-DR) were used for comprehensive investigation.
As described in the manuscript, the results showed that 1) the introduced Zone-X and Zone-DR radiomics driven sensing strategies significantly outperformed the traditional threshold driven strategy in terms of AUC. 2) the introduced Zone-DR strategy and Zone-SVM strategies achieved the highest sensitivity and specificity, respectively for ADC amongst the tested radiomics driven strategies, and 3) the introduced Zone-LR strategy achieved the highest sensitivity and specificity for CHB-DWI amongst the tested radiomics driven strategies. This work demonstrated that such a radiomic driven DWI sensing modality can provide a powerful strategy for zone-level prostate cancer sensing.
The study was designed and justified very well. Apparent diffusion coefficient (ADC) is the most commonly used DWI sensing modality currently. Computed high-b value diffusion weighted imaging (CHB-DWI) which is relatively new has not been widely adopted for cancer sensing. The study was carried out in a solid way based on results from 12,466 pathology-verified zones through the different DWI sensing modalities of 101 patients. Especially, the radiomics driven approaches (Zone-X and Zone-DR) have not been introduced previously for zone-level prostate cancer sensing.
Previous studies have focused on slice-level prostate cancer sensing. Zone-level cancer sensing with improved sensitivity and specificity can have enormous potential in clinic for prostate cancer screening, early detection, diagnosis, prognosis, surgical planning, and treatment evaluation.
The manuscript is organized and written well.
Suggestions:
- As shown in Table 3, the data are average values from many different samples. Therefore, it is important to describe information of the samples such as sample selection and number.
- It is necessary to show the related variability (standard deviations or standard errors) for the average values obtained.
- The data may be presented in an alternative way (such as a bar graph) for better comparison.
Reviewer 2 Report
Authors have presented classification architecture using CNN for Prostate cancer. However, I recommend them to do the following study to further enhance their work.
Visualize the class activation mapping results of the network in order to understand their network - https://arxiv.org/abs/1912.09621 Following work presents feature selection and clustering apporoaches in order to apply feature based classification approaches such as SVM which could be implemented by authors for their study - Narayanan, Barath Narayanan, Russell C. Hardie, Temesguen M. Kebede, and Matthew J. Sprague. "Optimized feature selection-based clustering approach for computer-aided detection of lung nodules in different modalities." Pattern Analysis and Applications 22, no. 2 (2019): 559-571. Narayanan, Barath Narayanan, Russell C. Hardie, and Temesguen M. Kebede. "Performance Analysis of Feature Selection Techniques for Support Vector Machine and its Application for Lung Nodule Detection." In NAECON 2018-IEEE National Aerospace and Electronics Conference, pp. 262-266. IEEE, 2018
Authors present machine learning approaches for Prostate cancer screening. They adopt feature based approach for classification. However, there are many feature-
based approaches for medical imaging that have proven to be successful as mentioned in the references in my review are pending, if the authors could adopt that approach, it will significantly improve their results. Authors use Support vector machine for classification, there is a paper which I had mentioned which specifically used SVM and feature selection methods for classification in medical imaging applications which could be adopted in this scenario, this would in turn increase the strength of the paper and would be of valuable interest to the readers. In addition, the authors present results using CNN, whose results can be visualized using class activation mapping results which is one of the papers I mentioned in my review. This would provide insight to the people who are reading author's work and would also understand how the CNN is classifying prostate cancer screening.
Round 2
Reviewer 2 Report
Authors have addressed my concerns and should be good to publish now.